# 3D-Structured and Blood-Contact-Safe Graphene Materials

**DOI:** 10.3390/ijms24043576

**Published:** 2023-02-10

**Authors:** Beata Kaczmarek-Szczepańska, Marta Michalska-Sionkowska, Pawel Binkowski, Jerzy P. Lukaszewicz, Piotr Kamedulski

**Affiliations:** 1Department of Biomaterials and Cosmetics Chemistry, Faculty of Chemistry, Nicolaus Copernicus University in Torun, Gagarina 7, 87-100 Torun, Poland; 2Department of Environmental Microbiology and Biotechnology, Faculty of Biology and Veterinary Sciences, Nicolaus Copernicus University in Torun, Lwowska 1, 87-100 Torun, Poland; 3Faculty of Chemistry, Nicolaus Copernicus University in Torun, Gagarina 7, 87-100 Torun, Poland; 4Centre for Modern Interdisciplinary Technologies, Nicolaus Copernicus University in Torun, Wilenska 4, 87-100 Torun, Poland

**Keywords:** 3D graphene, TEM, antioxidation, blood compatibility, carbon materials, graphene-based materials

## Abstract

Graphene is a promising material that may be potentially used in biomedical applications, mainly for drug delivery applications. In our study, we propose an inexpensive 3D graphene preparation method by wet chemical exfoliation. The morphology of the graphene was studied by SEM and HRTEM. Moreover, the volumetric elemental composition (C, N, and H) of the materials was analyzed, and Raman spectra of prepared graphene samples were obtained. X-ray photoelectron spectroscopy, relevant isotherms, and specific surface area were measured. Survey spectra and micropore volume calculations were made. In addition, the antioxidant activity and hemolysis rate in contact with blood were determined. Activity against free radicals of graphene samples before and after thermal modification was tested using the DPPH method. The RSA of the material increased after graphene modification, which suggests that antioxidant properties were improved. All tested graphene samples caused hemolysis in the range of 0.28–0.64%. The results showed that all tested 3D graphene samples might be classified as nonhemolytic.

## 1. Introduction

Nanomedicine is a new generation of medicine that involves the application of nanoscaled materials. Among them, graphene is an emerging material due to the variety of its properties. In most research, graphene has been studied in electrochemical applications as an electrode material for constructing supercapacitors, photovoltaic cells, and metal–air batteries [1,2,3,4,5,6]. Additionally, the co-authors of this paper contributed to this research trend [7,8,9,10]. Electrochemical tests showed the great potential of the obtained materials for the above-mentioned applications. Despite the progress regarding the electrochemical applications of graphene, some areas of potential applicability are still underestimated. The high chemical stability and relatively low reactivity of graphene let us hope for applications requiring such features [11]. In particular, the properties of graphene may be useful in cases when a chemically and biologically fragile matter may be deteriorated by contacting a reactive phase. This particularly applies to bio-originated matter and substances, which are commonly regarded as unstable and susceptible to damages of different origins.

By definition, graphene is a paternal 2D material; however, its 3D structuring is possible and may deliver a new sort of material comprising graphene’s unique properties (chemical stability, high electric conductivity, and mechanical strength) with ones typically attributed to standard 3D materials (porosity and enhanced surface area) [12,13,14].

The 2D nature of pristine graphene hinders its applicability to the immobilization of large molecules (drugs, enzymes, etc.) and microscopic objects such as cells or bacteria [15,16]. Such applications, in plenty of cases, rely on the presence of adequately developed pore structures due to increased adsorption potential inside the pores. Thus, any method which restructures the pristine 2D graphene flakes into a 3D porous structure is in favor of such applications. Three-dimensional graphene may form an environment appropriate for cell attachment, proliferation, and differentiation to its unique properties [17]. Theoretically, 3D graphene is a porous structure that provides good mechanical support for growing cells and tissue. It allows for rapid mass and electron transport necessary for the cell differentiation process [18]. Graphene application in nanomedicine provides promising solutions to present problems, such as the low mechanical properties of scaffolds, swelling behavior, which inhibits mass and electron transport, as well as high surface free energy, which limits cell attachment. Even if successful from the structural point of view, 3D structuring of graphene flakes should deliver a porous matrix of high compatibility to bio-originated matter such as blood cells, enzymes, etc. The current report tries to answer if the already discovered 3D structuring methods of commercial graphene may yield a porous matrix being somehow “friendly” to blood cells. This effect might be crucial for the application of 3D graphene matrixes as scaffolds for direct drug delivery to human blood.

Graphene has also been studied as an additive to polymeric matrixes. Such hybrid materials have shown improved mechanical properties [19]. Graphene oxide has been used as an additive for electrospun nanocomposite polycaprolactone (PCL)-based scaffolds. It improves osteogenic differentiation, which may enhance bone regeneration [20]. Graphene’s effectiveness is related to its ability to form different interactions (electrostatic, hydrogen binding, etc.) with biomolecules that induce cell differentiation. Additionally, it stimulates mechanosensing pathways, electrical activity, and cytoskeletal rearrangement [21]. Graphene-based nanosystems have been studied for anticancer therapy in breast cancer treatment. This provides promising graphene systems to replace traditional chemotherapy and radiotherapy [22,23]. There is a need to improve graphene properties using the wet chemical exfoliation method [14].

However, all modifications should be defined as safe and non-toxic. The aim of the proposed study was to determine graphene properties in terms of its applicability as a material for living-cell-bearing supports of advanced 3D architecture. The scheme of the method used by the authors is shown in Figure 1.

In such a case, non-destructive activity towards basic body fluids such as blood and lymph is absolutely essential since such cell-containing structures will stay in permanent contact with fluids. Moreover, the antioxidant properties of graphene-based materials should be tested, since this feature is a part of widely understood blood compatibility. In this work, the so-far-achieved goals in the 3D structuring of graphene flakes will be exploited to receive 3D graphene derivatives, the blood compatibility of which is still unknown. We assume that blood compatibility should be investigated since our 3D structuring protocol involves the addition of a hard template, a cationic surfactant, and other chemical reagents, which have to be used to achieve advanced pore structure. The up-to-date collected knowledge and skill by authors in the area of graphene exfoliation and subsequent 3D structuring has been assumed as a starting point for the design of 3D graphene derivatives that are safe in contact with blood.

## 2. Results and Discussion

Graphene and other carbon materials need to be characterized by a set of widely approved experimental methods, as in this study. Such features as surface area, pore structure, and chemical composition are crucial for any carbon-based material apart from biological activity.

The elemental composition and surface parameters for the obtained samples of porous 3D graphene are given in Table 1. The carbon content in all investigated samples is very high (higher than 90 wt.%). High carbon content is, among others, a measure of the oxidation of graphene after the performed 3D structuring. Potentially, chemical manipulations may lead to the oxidation of graphene and its conversion to graphene oxide. The aim of our study is to investigate the biocompatibility of graphene and not of graphene oxide. Thus, the carbon content above 90% is characteristic of materials accounted to the graphene category. This statement applies to all samples under investigation. There is a noticeable relationship between the degree of graphitization and the carbonization temperature range. Elemental analysis revealed that carbon content increases with increasing carbonization temperature in obtained materials. The results obtained in the case of the wet chemical exfoliation method indicated that Na_2_CO_3_ nanocrystals precipitate among exfoliated graphene flakes upon drying the reaction mixture. Thus, the template particles were removed by HCl etching and washing with distilled water. N_2_ sorption analysis showed that the specific surface area decreased (330–391 m^2^/g) below the value determined for the used paternal graphene nanoplatelets (750 m^2^/g). In parallel, changes of the pore structure became visible. The total pore volume decreased in all samples from 0.999 (GF-750) to 0.220 cm^3^/g (GF_Na_600). Surprisingly, the contribution of the mesopore volume V_me_ to the total pore volume V_t_ increased from 36% to 79%. It is notable that for three of four investigated samples the share of mesopores is close to 50% or much higher. The presence of mesopores (average pore diameter from 2 to 50 nm) is significant for applications involving a liquid environment where adsorption/desorption proceeds (for example, drug delivery).

Additionally, the morphology details of representative samples are shown in the HRTEM images (Figure 2). Figure 2 shows that all investigated 3D graphene samples had similar irregular surface characteristics due to randomly oriented graphene layers. Mainly, HRTEM allowed thin graphene flakes to be observed. It has to be stated that, in general, SEM morphology is not informative and therefore, HRTEM images need to be acquired. Two carbon phases are visible on the HRTEM images (Figure 2): exfoliated graphene sheets and bundle-looking domains of amorphous carbon, which is typical of a carbon phase originating from the thermal decomposition of polyfurfuryl alcohol applied for a durable stacking of graphene flakes.

In the case of carbon-based materials, the surface elemental composition may substantially differ from the bulk elemental content. In general, carbon is a reactive element, and surface atoms may come into reaction with gases present in the atmosphere. In particular, oxygen is supposed to yield diversified surface-oxygen-based species. Therefore, two methods need to be applied to determine elemental composition: elemental combustion analysis (bulk analysis) and XPS (surface analysis).

The results of combustion elemental analysis only allowed the bulk content of three elements to be determined: C, N, and H, while the specific chemical environment of the heteroatoms embedded in the carbon matrix remains unknown. Therefore, obtained samples were additionally supported by X-ray photoelectron spectroscopy (XPS) investigations. A sample with the most promising biological properties was selected for the XPS study based on DPPH and blood compatibility. The XPS spectra of the representative GF_Na_800 sample were determined and are demonstrated in Figure 3 and Table 2.

The XPS elemental content of carbon was high (88.6 at.%) and very close to the values obtained by combustion analysis. However, XPS C content is slightly lower, which is a natural effect of spontaneous oxidation of the carbon surface by atmospheric oxygen. Carbon atoms were mostly bonded as sp^3^ hybridized atoms (band C 1s at binding energy 285.0 eV), which is characteristic of graphene materials. The C 1s spectra of the GF_Na_800 sample are composed of four peaks corresponding to C-C bond (sp^3^) peak at 285 eV [24]; C-O-C or C-OH or C-NH bond peak at 286.3 eV [25]; C=O or O-C-O or N-C-O bond peak at 287.7 eV [26]; and O-C=O peak at 288.6 eV [26]. The total amount of oxygen is in a range from 9.6 at.%. The peak at 532.0 eV signifies the presence of a O-C-N or C=O bond and the peak at 533.3 eV is characteristic of a O*=C-O or O-C-O bond [24,26,27]. Moreover, the presence of nitrogen is noticeable (1.8 at.%). Nitrogen was used only as an additive surfactant and during the carbonization process as an inert gas flow. The high-resolution N 1s spectra can be deconvoluted into one peak, located at 400.5 eV, which is attributed to quaternary (N-Q) groups [28].

In turn, the Raman studies confirm exfoliation to FLG as a building block of the final structure [9,29]. Figure 4 and Table 3 contain information on the D, G, and 2D band placement and intensity.

The intensity ratios I_2D_/I_G_ and I_D_/I_G_ prove that the investigated samples contain mainly multilayered graphene. The carbonization temperature of the presented samples affects the position of the G and 2D bands. The GF_Na_900 sample had more vacancies and disorders since it had the highest I_D_/I_G_ ratio and moreover the highest carbon at.%.

The activity against free radicals of graphene samples before and after thermal modification was tested using the DPPH method (Table 4). The RSA of the material increased after graphene modification, which suggests that antioxidant properties are improved. Graphene with the 750 m^2^/g area showed higher RSA than 300 m^2^/g. Additionally, with the increase in the modification temperature, the RSA increases. The maximum RSA = 89% was noticed after graphene treatment at 800 °C. When 900 °C temperature was applied, RSA decreased rapidly.

The material compatibility with blood is an important factor that classifies its potential medical application. Materials that are implanted in the body may not show hemolysis above 5% [30]. As it is listed in Table 5, all tested graphene samples caused hemolysis in the range of 0.28–0.64%. The increase in the temperature in the preparation process results in an increase in the hemolysis rate.

The authors have performed multidirectional and extended studies on the synthesis of 3D graphene as a porous material in recent years. Synthesis methodology has been described in several papers, among which some of the most meaningful may be cited [8,9,14]. The aims of nearly all performed studies were exclusively focused on energy applications, including graphene-based electrode materials for batteries, supercapacitors, and photovoltaic cells. However, the well-developed pore structure and high carbon purity of such materials have encouraged authors to find radically new applications as a porous platform for bioactive species and/or in permanent contact with constituents of living organisms. Several investigations were performed to define some bioapplicability limitations of 3D structured graphene platforms with respect to the mentioned target.

Antioxidants protect cells against injury by free radicals. The antioxidant activity of materials introduced to the human body is important to protect cells during medical treatment [31]. Graphene-based materials have been characterized as antioxidants by Qiu et al. [32]. Radical scavenging activity was inversely proportional to their total surface area. However, our modification method results in an increase in surface area as well as antioxidant activity. The modification method is a crucial method for safe and effective graphene-based material preparation. Hemocompatibility is one of the crucial parameters to consider with materials for potential biomedical applications. Erythrocytes are sensitive to hemolysis due to shear stress. Insufficient hemocompatibility has been found to impair safety through the activation of blood coagulation [33]. We compared our results with ASTM F756-00 standard materials where the hemolytic index 0–2% is considered as non-hemolytic, 2–5% is slightly hemolytic, and <5% is classified as hemolytic [34]. We observed that all tested graphene samples might be classified as non-hemolytic. Thereby, they are allowed to be proposed for biomedical application as they have antioxidant activity as well as low rates of hemolysis.

## 3. Methods and Materials

### 3.1. The Synthesis of 3D Graphene

Commercial graphene-type precursors, i.e., nanoplatelets (Sigma Aldrich, Poznan, Poland) were modified to three-dimensional graphene using the original self-authored method. Two types of graphene nanoplatelets were applied with respect to their surface area: 300 m^2^/g and 750 m^2^/g. After preliminary tests, only graphene flakes with an area of 750 m^2^/g were finally selected. The self-authored method is based on the application of a hard template. Nanopowder of Na_2_CO_3_ (POCh, Gliwice, Poland) was chosen as a templated for further experiments. The formerly elaborated manufacturing protocols were applied [9,14]. Firstly, the aqueous solution of Na_2_CO_3_ (16.9%) was supplemented with 2 g of raw graphene nanoplatelets and then mixed for 1 h in a magnetic stirrer at 20 °C. Next, 20 mL of 1-methyl-2-pyrrolioline (Sigma-Aldrich, Poznan, Poland) was added, mixed with 0.1 mL of a CTAC (Sigma-Aldrich, Poznan, Poland) solution 25 wt.% in H_2_O, and then mixed well for 1 h in a magnetic stirrer [14]. After this time had elapsed, the flask was placed for one hour in an ultrasonic bath (at room temperature, 40 Hz frequency, and 100 watt power), and washed with distilled water using a Büchner funnel as in our previous work [14]. Next, the samples under elaboration needed to by dried (in electric furnace at 80–100 °C for 20 h) and diminished (by grounding in a mortar). Every 1 g of powder was mixed with 10 mL of furfuryl alcohol, and then 1 drop of concentrated phosphoric acid aqueous solutions (75%) was added. The samples were heated at 80 °C in an oven for 24 h to allow for the polycondensation of furfuryl alcohol. The samples were subsequently carbonized at temperatures of 600 to 900 °C for 1 h to 1.5 h as in our previous work [14]. The prepared material was heated under a flow of N atoms at a rate of 10 °C/min in a furnace (Thermolyne F21100, NIST, Gaithersbur, MD, USA). After carbonization process, the samples were treated with concentrated (34–37%) HCl for 25 min, and next materials were rinsed with distilled water using a Büchner funnel until the pH was 6–7 as in our previous work [9,14]. Then, the material was dried in an electric furnace at 105 °C for 24 h. The etching with HCl allowed pores to open in the matrix. The 3D graphene obtained in these methods was denoted as GF_Na_T, where GF_Na means the mass ratio of graphene nanoplatelets (2 g) to Na_2_CO_3_ (6 g), T is the carbonization temperature (°C). GF_Na_raw means the sample after modifications with Na_2_CO_3_ but before carbonization. The sample denoted as GNP_300 and GNP_750 means pure graphene nanoplatelets with surface areas 300 m^2^/g or 750 m^2^/g used in the experiments, respectively. The only losses were the amounts that were retained on the wall of the laboratory vessels. To minimize carbon material losses, all laboratory glass equipment were washed with distilled water, and the residues of material were added to its main portion as in our previous works [9,14]. Therefore, the weight losses were lower than 5%, and the yield can be estimated as about 95% in relation to the raw material.

### 3.2. Materials Characterization

The morphology of the synthetized 3D graphene samples was determined by high-resolution transmission electron microscopy (HRTEM, FEI Europe production, model Tecnai F20 X-Twin, Brno, Czech Republic). The carbons obtained prior to the HRTEM analysis were dispersed in ethanol as a solvent and treated with an Inter Sonic IS-1K bath for 20 min and deposited on holey carbon-coated copper grids [9,14]. The elemental composition was analyzed by a combustion elemental analyzer (Vario MACRO CHN, Elementar Analysensysteme GmbH, Langenselbold, Germany) as described formerly [9,14]. Raman spectra were recorded with a micro-Raman spectrometer (532 nm laser wavelength, Senterra, Bruker Optik, Billerica, MA, USA). To focus the laser beam, a 50× microscope objective was used. An excitation power of 2 mW was found to be optimal regarding the sample stability. The following settings of Raman spectrometer were applied: the resolution 4 cm^−1^, CCD temperature 223 K, laser spot diameter 2.0 µm, and total integration time 100 s (50 × 2 s) as described formerly [9,14]. X-ray photoelectron spectroscopy (XPS, PHI5000 VersaProbe II Scanning XPS Microprobe, Chigasaki, Japan) measurements were performed using a monochromatic Al Kα X-ray source. Survey spectra were recorded for all materials in the energy range of 0 to 1300 eV with a 0.5 eV step; spectra were recorded with a 0.1 eV step [9]. Additionally, all synthetized samples were tested by means of the low-temperature adsorption of nitrogen method at −196 °C (ASAP 2010 Micromeritics, Norcross, GA, USA). Measurement preceded the vacuum outgassing at 200 °C for 2 h [9,14]. The specific surface area (S_BET_) was determined based on the Brunauer–Emmett–Teller (BET) method from nitrogen adsorption data in a relative pressure range of 0.02 to 0.2. The micropore volume (V_mi_) was calculated using the t-plot method by MicroActive program (Micromeritics, Norcross, GA, USA). The total pole volume (V_t_) was determined from the amount of gas adsorbed at a relative pressure of 0.99 [9,14].

### 3.3. DPPH Radical Scavenging Assay

The DPPH (2,2-Diphenyl-1-picrylhydrazyl, free radical, 95%; Alfa Aesar, Lancashire, UK) radical scavenging assay allows the antioxidant properties to be determined [35]. The 2 mL of DPPH solution (250 µM in methyl alcohol) was added to 1 g of each sample (n = 5) and placed in a 24-well plate. Samples were stored in the darkness for 1 h. The absorbance was measured spectrophotometrically at 517 nm (UV-1800, Shimadzu, Switzerland). The RSA was calculated with the following equation:(1)$$\mathrm{RSA}\%=\frac{{\mathrm{Abs}}_{\mathrm{DPPH}}-{\mathrm{Abs}}_{\mathrm{PB}}}{{\mathrm{Abs}}_{\mathrm{DPPH}}}*100\%$$
where
Abs_DPPH_ is the absorbance of the fresh DPPH solution;Abs_PB_ is the absorbance of the fresh DPPH solution after contact with tested samples.

### 3.4. Blood Compatibility

For the study of compatibility of the graphene samples with blood, anticoagulated sheep blood (0.2 mL) was added to physiological saline solution (10 mL) which contained 1 g of each sample (n = 3) according to ASTM Hemolysis (Direct Contact and Extract Methods) GLP Report. Two control samples with the addition of fresh blood were prepared: positive (in water) and negative (in physiological saline). The mixture was incubated (37 °C, 1 h) and the suspension was centrifuged (at 1000 rpm, 10 min). The absorbance was measured spectrophotometrically at 540 nm by using the Multiscan FC (Thermo Fisher Scientific, Waltham, MA, USA) [36]. Finally, the hemolysis rate was then calculated using the following equation:(2)$$\mathrm{rate}\;\mathrm{of}\;\mathrm{hemolysis}\;\left[\%\right]=\frac{\left[\mathrm{OD}\right]\mathrm{specimen}-\left[\mathrm{OD}\right]\mathrm{negative}}{\left[\mathrm{OD}\right]\mathrm{positive}-\left[\mathrm{OD}\right]\mathrm{negative}}*100\%$$
where
[OD] specimen is the absorbance of solution in contact with tested samples;[OD] negative is the absorbance of blood in physiological saline;[OD] positive is the absorbance of blood in water.


## 4. Conclusions

The applied manufacturing protocol, despite some chemical reagents used as hard templates and cationic surfactants, delivers a blood-compatible porous material potentially applicable for further living-cell-involved manipulations. The antioxidant activity of the material increased after graphene modification. The maximum RSA = 89% was noticed after graphene treatment at 800 °C. Moreover, the increase in the temperature in the preparation process results in an increase in the hemolysis rate. The structural properties indicate a high quality of obtained porous graphene structures with a well-developed surface area and pore structure, which is essential for living cell immobilization. We have proven that the manufacturing protocol is relatively environmentally friendly and low-cost. These aspects of the performed study will be continued in further research.

## Figures and Tables

**Figure 1 ijms-24-03576-f001:**
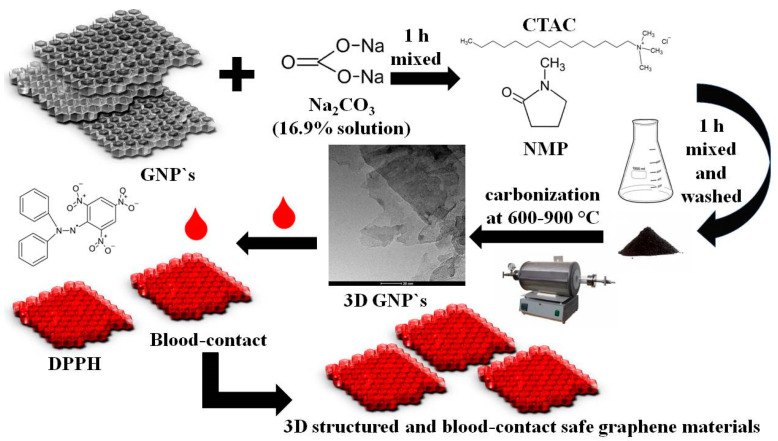
The proposed method of synthesis of blood-contact-safe 3D graphene materials.

**Figure 2 ijms-24-03576-f002:**
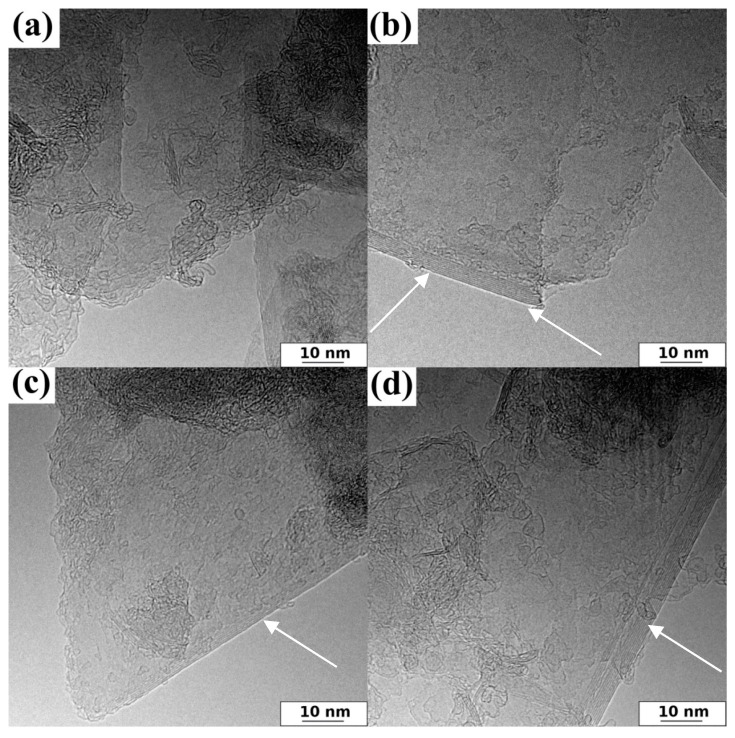
Representative HRTEM images of obtained samples: (**a**) GF_Na_600 sample, (**b**) GF_Na_700 sample, (**c**) GF_Na_800 sample, and (**d**) GF_Na_900 sample, (white arrows on the image (**b**–**d**) showing pure and thin sheets).

**Figure 3 ijms-24-03576-f003:**
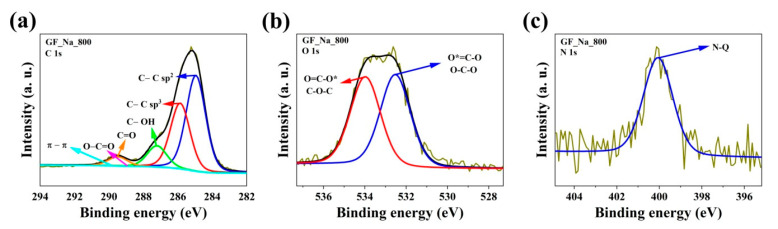
High-resolution X-ray photoelectron spectra for C1s, O1s, and N1s of representative sample: (**a**) C1s of GF_Na_800, (**b**) O1s of GF_Na_800, and (**c**) N1s of GF_Na_800.

**Figure 4 ijms-24-03576-f004:**
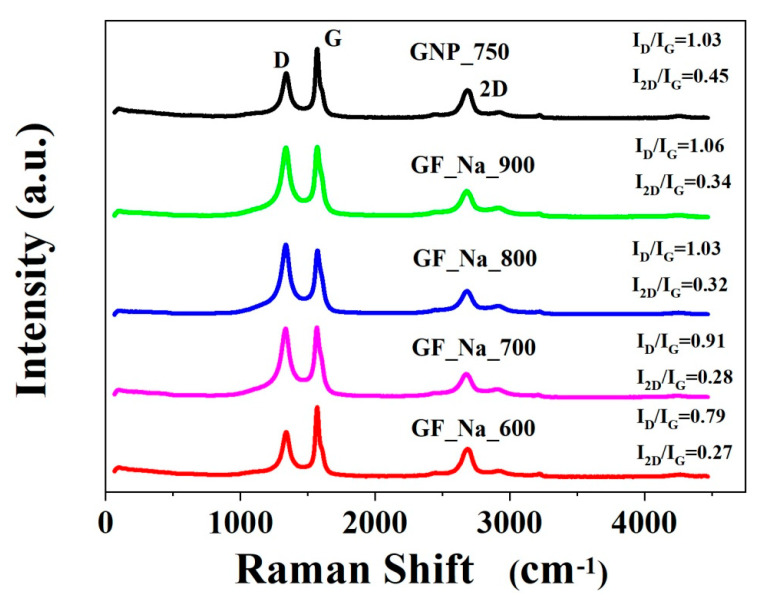
Raman spectra of obtained materials.

**Table 1 ijms-24-03576-t001:** Elemental composition and porosity parameters obtained from N_2_ sorption analysis of 3D graphene.

Sample	Elemental Analysis (wt.%)			S_BET_ (m^2^/g)	V_t_ (cm^3^/g)	V_mi_ (cm^3^/g)	V_me_ (cm^3^/g)	V_me_/V_t_ × 100%
	C	N	H					
GNP_750	87.3	0.7	0.9	750	0.999	0.127	0.872	87%
GF_Na_600	90.9	0.4	0.8	330	0.220	0.141	0.079	36%
GF_Na_700	91.6	0.5	1.0	332	0.248	0.133	0.115	46%
GF_Na_800	92.9	0.4	0.7	357	0.279	0.127	0.152	54%
GF_Na_900	93.7	0.6	0.9	391	0.377	0.078	0.299	79%

**Table 2 ijms-24-03576-t002:** The elemental composition of the obtained representative GF_Na_800 sample using XPS measurements.

Element	C					O			N
Binding energy (eV)	285.0	286.3	287.7	288.6	% of total	532.0	533.3	% of total	400.5
Sample	Content (at.%)					Content (at.%)			Content (at.%)
GF_Na_800	44.2	30.3	9.8	4.2	88.6	4.9	4.7	9.6	1.8

**Table 3 ijms-24-03576-t003:** Results of Raman spectra and D, G, and 2D band data for obtained materials.

Sample	I_D_	cm^−1^	I_G_	cm^−1^	I_2D_	cm^−1^	I_D_/I_G_	I_2D_/I_G_
GNP_750	1.00	1344.5	0.98	1579.0	0.44	2679.0	1.03	0.45
GF_Na_600	0.99	1337.0	1.00	1592.0	0.38	2703.5	0.79	0.27
GF_Na_700	0.99	1345.5	1.00	1587.0	0.37	2695.0	0.91	0.28
GF_Na_800	1.00	1343.0	1.00	1584.0	0.37	2699.0	1.03	0.32
GF_Na_900	1.00	1343.0	1.00	1582.0	0.36	2687.0	1.06	0.34

**Table 4 ijms-24-03576-t004:** Antioxidant activity (RSA%) of graphene samples.

Specimen	RSA%
GNP_300	77.61 ± 0.12
GNP_750	84.72 ± 1.11
GF_Na_raw	57.92 ± 0.44
GF_Na_600	80.51 ± 2.96
GF_Na_700	83.93 ± 0.75
GF_Na_800	89.00 ± 0.21
GF_Na_900	71.85 ± 0.63

**Table 5 ijms-24-03576-t005:** The hemolysis rate [%] for different types of graphene samples.

Specimen	Rate of Hemolysis [%]
GNP_300	0.29 ± 0.21
GNP_750	0.49 ± 0.25
GF_Na_raw	0.28 ± 0.08
GF_Na_600	0.50 ± 0.11
GF_Na_700	0.53 ± 0.10
GF_Na_800	0.58 ± 0.33
GF_Na_900	0.64 ± 0.08

## Data Availability

The data presented in this study are available on request from the corresponding authors.

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
