# Peer review of "3D-Structured and Blood-Contact-Safe Graphene Materials"

_ijms, 2023, doi:10.3390/ijms24043576_

Round 1
Reviewer 1 Report
The manuscript present interesting data on the 3D structured and blood-contact safe graphene materials. However, the flow structure of the manuscript is not interesting the way it is presented from point of view. I would like the authors to change the flow to: Introduction, Experimental and materials characterization followed by Results and discussion and end with conclusion.
Author Response
On behalf of all authors, I would like to express our gratitude for a careful reading of our manuscript and subsequent constructive remarks. We intend to improve our manuscript in accordance with these comments. Please see our point-to-point responses to the reviewers’ comments, marked in red below. The corresponding changes to the comments are highlighted with a yellow background in the revision.
Sincerely yours,
PhD Piotr Kamedulski, corresponding author
REVIEWER 1
The manuscript present interesting data on the 3D structured and blood-contact safe graphene materials. However, the flow structure of the manuscript is not interesting the way it is presented from point of view. I would like the authors to change the flow to: Introduction, Experimental and materials characterization followed by Results and discussion and end with conclusion.
Response: We thank the Reviewer for this remark. We changed the structure of the manuscript according to the reviewer’s instructions.

Reviewer 2 Report
Authors reported nonhemolytic properties of the prepared 3D graphene. There are already similar reports, so I recommend to reject the paper. Please resubmit after addressing following issues:
1. Improve introduction. Add previous studies and explain drawbacks and ways to improve.
2. iThenticate shows that there are 40% similarities. Please re-write those parts. There are many places where there are word-to-word copy-paste.
3. Better explain novelty of your work.
Author Response

(The authors gave the same response as above.)

Reviewer 3 Report
1. Writing need be improved.
2. Temperature were used as the adjustable factor, but the effect of temperature is not stated in the discussion or conclusion.
3. Conclusion does not reflect the result and discussion. Does modification improve the antioxidation and hemocaompatibility the goal of the study
Author Response
On behalf of all authors, I would like to express our gratitude for a careful reading of our manuscript and subsequent constructive remarks. We intend to improve our manuscript in accordance with these comments. Please see our point-to-point responses to the reviewers’ comments, marked in red below. The corresponding changes to the comments are highlighted with a yellow background in the revision.
Sincerely yours,
PhD Piotr Kamedulski, corresponding author
REVIEWER 3
- Writing need be improved.
Response: We thank the Reviewer for this remark. Once again we have carefully checked and improved the English writing in the revised manuscript.
- Temperature were used as the adjustable factor, but the effect of temperature is not stated in the discussion or conclusion.
Response: We thank the Reviewer for this remark. We used temperature range 600-900°C. We write in the manuscript “Also, with the increase of the modification temperature, the RSA increases. The maximum RSA=89% was noticed after graphene treatment in 800°C. When 900°C temperature was applied, RSA decreased rapidly.”
- Conclusion does not reflect the result and discussion. Does modification improve the antioxidation and hemocaompatibility the goal of the study.
Response: We thank the Reviewer for this remark. Conclusion section is now improved. We add sentences: The antioxidant activity of material increased after graphene modification. The maximum RSA=89% was noticed after graphene treatment in 800°C. Moreover, the increase in the temperature in the preparation process results in an increase in the hemolysis rate.

Reviewer 4 Report
In this paper, the authors propose an inexpensive 3D graphene preparation method by wet chemical exfoliation, blood-compatible and potentially useful as a support for cell growth. Although the study is interesting, some issues need to be resolved or improved before it can be published.
Major concern
1. The authors mention their contributions to graphene multiple times in the introduction. I suggest that these contributions be described (at least briefly) so that they can be linked to the current manuscript. Lines 166-171 (3. Discussion) could, for example, be expanded in the introduction.
The nanotechnology applications of graphene are discussed briefly. As a result, the authors must stress the significance of this material in 3D form, with specific applications.
What kinds of cells might this support be used for if the end goal is to employ 3D graphene as a support for cell growth? What is the final application? Cellular regeneration, tissue regeneration?...
All of these issues must be addressed in the introduction so that the manuscript is more robust.
I suggest that the authors add a figure that depicts the proposed technique for synthesize the graphene structure proposed in this manuscript.
2. Section 2 "results" requires the authors to thoroughly explain the differences between each of the samples tested. I suppose GNP_750 is the control and 750 is the surface area; what is the difference between the remaining samples? Please elaborate on this point, as well as the results in Table 1.
3. According to the data in Table 1, the results of the GF_Na_600 and GF_Na_700 samples are very similar; however, there is a greater variation in the area and volume with respect to GF_Na_ 900; what is this behavior due to, if there is no significant difference in the structure between the four samples studied in the HRTEM images?
It could be more advisable to study the morphology of the proposed structure using the FE-SEM technic.
4. To determine the stability of the graphene samples in water and PBS pH 7.4, I advise the authors to analyze the zeta potential of the samples.
5. Why was the GF_Na_800 sample the only one picked for XPS analysis? Why wasn't this method used to evaluate the remaining samples? This point should be clarified in the manuscript.
6. Why was graphene GNP_300 not characterized like the rest of the samples? Why wasn't the GF_Na_raw sample previously characterized and what does it correspond to?
7. 1 mg of the graphene sample was used to test its antioxidant capacity, and 1 g was utilized to test its blood compatibility, so, what is the cause of these large differences? The amount of graphene sample chosen by the authors must be justify.
Other graphene amount than 1g were explored to examine blood compatibility.
8. The results in Table 5 could be supported by optical microscope images, because lysed red blood cells are plainly visible.
9. The discussion of results appears to be more like the manuscript's conclusions. The authors should expand on this section by emphasizing on the research findings.
Because the authors compare the results of their samples with the graphene standard ASTM F756-00, the physicochemical properties of this material should be presented in section 3 of this manuscript.
10. Again, if the authors' ultimate goal is to employ 3D graphene as a support for cell growth, they should conduct in-vitro cell growth tests on the graphene structure. It is also recommended that a live-dead assay be performed to verify cell viability on the support or cytotoxicity assays, as one of the key disadvantages of graphene-based nanosystems is their cytotoxicity.
Minor concern
1. Bibliographic references must be provided to back up the affirmation of lines 133-136.
2. The statement “Materials that are implanted in the body may not show 158 hemolysis above 5%.” Line 158-159 must be justified with an adequate bibliographical reference.
3. Due to the number of acronyms used in this manuscript, authors must include a list of abbreviations.

Author Response
On behalf of all authors, I would like to express our gratitude for a careful reading of our manuscript and subsequent constructive remarks. We intend to improve our manuscript in accordance with these comments. Please see our point-to-point responses to the reviewers’ comments, marked in red below. The corresponding changes to the comments are highlighted with a yellow background in the revision.
Sincerely yours,
PhD Piotr Kamedulski, corresponding author
REVIEWER 4
In this paper, the authors propose an inexpensive 3D graphene preparation method by wet chemical exfoliation, blood-compatible and potentially useful as a support for cell growth. Although the study is interesting, some issues need to be resolved or improved before it can be published.
Major concern
- The authors mention their contributions to graphene multiple times in the introduction. I suggest that these contributions be described (at least briefly) so that they can be linked to the current manuscript. Lines 166-171 (3. Discussion) could, for example, be expanded in the introduction.
The nanotechnology applications of graphene are discussed briefly. As a result, the authors must stress the significance of this material in 3D form, with specific applications.
What kinds of cells might this support be used for if the end goal is to employ 3D graphene as a support for cell growth? What is the final application? Cellular regeneration, tissue regeneration?...
Response: We thank the Reviewer for this remark. We considered 3D graphene for bioimaging applications, drug and gene delivery. We have revised the introduction and the text after changing is highlighted in red color.
All of these issues must be addressed in the introduction so that the manuscript is more robust.
I suggest that the authors add a figure that depicts the proposed technique for synthesize the graphene structure proposed in this manuscript.
Response: We thank the Reviewer for this remark. We prepare a graphical abstract to show proposed technique and potential apllications:
- Section 2 "results" requires the authors to thoroughly explain the differences between each of the samples tested. I suppose GNP_750 is the control and 750 is the surface area; what is the difference between the remaining samples? Please elaborate on this point, as well as the results in Table 1.
Response: We thank the Reviewer for this remark. In section 2. Experimental and materials characterization/ 2.1. The synthesis of 3D graphene we explain the name of samples.
- According to the data in Table 1, the results of the GF_Na_600 and GF_Na_700 samples are very similar; however, there is a greater variation in the area and volume with respect to GF_Na_ 900; what is this behavior due to, if there is no significant difference in the structure between the four samples studied in the HRTEM images?
It could be more advisable to study the morphology of the proposed structure using the FE-SEM technic.
Response: We thank the Reviewer for this remark. The explanation is added to the section Results and discussion.
- To determine the stability of the graphene samples in water and PBS pH 7.4, I advise the authors to analyze the zeta potential of the samples.
Response: We thank the Reviewer for this remark. Unfortunately, we do not have such equipment at the Faculty of Chemistry NCU in Torun and it is difficult to add results until the deadline for submission of the revised manuscript. So analyze the zeta potential of the samples is not performed in this paper.
- Why was the GF_Na_800 sample the only one picked for XPS analysis? Why wasn't this method used to evaluate the remaining samples? This point should be clarified in the manuscript.
Response: We thank the Reviewer for this remark. XPS is an expensive method, the cost of one sample is about 1000 PLN. One sample was selected with the most promising biological properties.
- Why was graphene GNP_300 not characterized like the rest of the samples? Why wasn't the GF_Na_raw sample previously characterized and what does it correspond to?
Response: We thank the Reviewer for this remark. We have done preliminary research with different graphene nanoplatelets but for further modifications, graphene flakes with an area of 750 m2/g were finally selected.
- 1 mg of the graphene sample was used to test its antioxidant capacity, and 1 g was utilized to test its blood compatibility, so, what is the cause of these large differences? The amount of graphene sample chosen by the authors must be justify.
Other graphene amount than 1g were explored to examine blood compatibility.
Response: We thank the Reviewer for this remark. We apologize for the mistake, also 1g was taken for the antioxidant capacity studies. It is now corrected.
- The results in Table 5 could be supported by optical microscope images, because lysed red blood cells are plainly visible.
Response: We thank the Reviewer for this remark. We agree that lysed red blood cells are plainly visible. However, in our opinion it is important to consider the blood compatibility that is related to hemolysis rate. Hemolysis testing is the most common method to determine the hemocompatibility properties of biomaterials.
- The discussion of results appears to be more like the manuscript's conclusions. The authors should expand on this section by emphasizing on the research findings.
Because the authors compare the results of their samples with the graphene standard ASTM F756-00, the physicochemical properties of this material should be presented in section 3 of this manuscript.
Response: We thank the Reviewer for this remark. We have added several new lines to “Discussion”. The new text is marked in red. We cited the appropriate reference as ASTM F756-00 is a standard practice for assessment of hemolytic properties of materials. This practice provides a protocol that’s is assigned as ASTM F756-00.
- Again, if the authors' ultimate goal is to employ 3D graphene as a support for cell growth, they should conduct in-vitro cell growth tests on the graphene structure. It is also recommended that a live-dead assay be performed to verify cell viability on the support or cytotoxicity assays, as one of the key disadvantages of graphene-based nanosystems is their cytotoxicity.
Response: We thank the Reviewer for this remark. We dedicate the 3D graphene for medical application such as bioimaging applications, drug and gene delivery. We done the blood compatibility study to test the compatibility of graphene with blood. In our opinion material would not have direct contact with human cells.
Minor concern
- Bibliographic references must be provided to back up the affirmation of lines 133-136.
Response: We thank the Reviewer for this remark. New references are added in this lines.
- The statement “Materials that are implanted in the body may not show 158 hemolysis above 5%.” Line 158-159 must be justified with an adequate bibliographical reference.
Response: We thank the Reviewer for this remark. The appropriate reference is now added.
- Due to the number of acronyms used in this manuscript, authors must include a list of abbreviations.
Response: We thank the Reviewer for this remark. In section 2. Experimental and materials characterization/ 2.1. The synthesis of 3D graphene we explain the name of samples.

Round 2
Reviewer 2 Report
There is an improvement in the manuscript, but there are still issues with the paper.
1. I have run another iThenticate with the manuscript and it is still above 30%. It does not matter, If you are using the same methods or instruments, you need to re-word to reduce the similarity below 25%.
2. In the introduction part, especially newly added parts, needs references.
Author Response
On behalf of all authors, I would like to express our gratitude for a careful reading of our manuscript and subsequent constructive remarks. We intend to improve our manuscript in accordance with these comments. Please see our point-to-point responses to the reviewers’ comments, marked in red below. The corresponding changes to the comments are highlighted with a yellow background in the revision.
Sincerely yours,
PhD Piotr Kamedulski, corresponding author
Reviewer 2
There is an improvement in the manuscript, but there are still issues with the paper.
- I have run another iThenticate with the manuscript and it is still above 30%. It does not matter, If you are using the same methods or instruments, you need to re-word to reduce the similarity below 25%.
Response: We thank the Reviewer for this remark. We change the words again to reduce the similarity. Now is 9% in Grammarly but when we delete references then is 2%.
- In the introduction part, especially newly added parts, needs references.
Response: We thank the Reviewer for this remark. We add new references.

Reviewer 3 Report
I would suggest the author to futher editing the manuscript to increase its readability.
Author Response
On behalf of all authors, I would like to express our gratitude for a careful reading of our manuscript and subsequent constructive remarks. We intend to improve our manuscript in accordance with these comments. Please see our point-to-point responses to the reviewers’ comments, marked in red below. The corresponding changes to the comments are highlighted with a yellow background in the revision.
Sincerely yours,
PhD Piotr Kamedulski, corresponding author
Reviewer 3
I would suggest the author to futher editing the manuscript to increase its readability.
Response: We thank the Reviewer for this remark. We changed the structure of the manuscript and add new sentences, and references. Once again we have carefully checked and improved the English writing in the revised manuscript.

Reviewer 4 Report
After reviewing the corrected version of the manuscript, authors should address the following points.
1. Regardless of the graphical abstract (which is required), authors must add an explanatory figure in the introduction.
2. If the GF-Na-800 sample was chosen because it has better biological properties, this must be stated in the manuscript, as well as why it was chosen to proceed with the characterization.
3. If the samples GNP_300 and GF_Na_raw are mentioned by the authors in the manuscript, they must explain exactly what they are and why they haven't been characterized.
Additionally, if the authors had earlier characterizations, they might provide them to explain why a certain kind of graphene was ultimately chosen, as is the case with the sample GF_Na 800.
4. Other graphene amount than 1g were explored to examine blood compatibility. The prior review did not touch on this point. The selection of the amount of graphene must be justified in the manuscript.
5. I again advise that the authors include an acronyms list. Authors should make it simple for readers to understand their work, however in this case, the lack of an acronym list makes it difficult to comprehend.

Author Response
On behalf of all authors, I would like to express our gratitude for a careful reading of our manuscript and subsequent constructive remarks. We intend to improve our manuscript in accordance with these comments. Please see our point-to-point responses to the reviewers’ comments, marked in red below. The corresponding changes to the comments are highlighted with a yellow background in the revision.
Sincerely yours,
PhD Piotr Kamedulski, corresponding author
Reviewer 4
After reviewing the corrected version of the manuscript, authors should address the following points.
- Regardless of the graphical abstract (which is required), authors must add an explanatory figure in the introduction.
Response: We thank the Reviewer for this remark. We add new Figure 1 in section - Introduction.
- If the GF-Na-800 sample was chosen because it has better biological properties, this must be stated in the manuscript, as well as why it was chosen to proceed with the characterization.
Response: We thank the Reviewer for this remark. We add sentence: A sample with the most promising biological properties was selected for the XPS study based on DPPH and blood compatibility. The XPS spectra of the representative GF_Na_800 sample were determined and are demonstrated in Figure 3 and Table 2.
- If the samples GNP_300 and GF_Na_raw are mentioned by the authors in the manuscript, they must explain exactly what they are and why they haven't been characterized.
Additionally, if the authors had earlier characterizations, they might provide them to explain why a certain kind of graphene was ultimately chosen, as is the case with the sample GF_Na 800.
Response: We thank the Reviewer for this remark. We add before introduction an acronyms list. Moreover, in section 2. Experimental and materials characterization/ 2.1. The synthesis of 3D graphene we explain all the name of samples too. Commercial raw graphene nanoplatelets (Sigma Aldrich, Poznan, Poland) were modified to three-dimensional graphene using the original self-designed method. Two series of graphene nanoplatelets differing in surface area (300 m2/g and 750 m2/g) were selected for the preliminary tests. DPPH and blood compatibility testing for GNP_300 sample showed the worst parameters. However, for further modifications, graphene flakes with an area of 750 m2/g were finally selected. GF_Na_raw – means the sample after modifications with Na2CO3 but before carbonization.
- Other graphene amount than 1g were explored to examine blood compatibility. The prior review did not touch on this point. The selection of the amount of graphene must be justified in the manuscript.
Response: We thank the Reviewer for this remark. The amount of graphene form the hemolysis rate studies was determined according ASTM Hemolysis (Direct Contact and Extract Methods) GLP Report where the concentration of studied materials is 0.1 g/ml. As we added the 10mL od blood containing solution, the amount of graphene used for the study was determined as 1g.
We add sentence in the manuscript: For the study of compatibility of the graphene samples with blood, anticoagulated sheep blood (0.2 mL) was added to physiological saline solution (10 mL) which contained 1g of each sample (n=3) according to ASTM Hemolysis (Direct Contact and Extract Methods) GLP Report.
- I again advise that the authors include an acronyms list. Authors should make it simple for readers to understand their work, however in this case, the lack of an acronym list makes it difficult to comprehend.
Response: We thank the Reviewer for this remark. We add before introduction an acronyms list.

Round 3
Reviewer 4 Report
I have no further comments.